# An Insight into Symmetrical Cyanine Dyes as Promising Selective Antiproliferative Agents in Caco-2 Colorectal Cancer Cells

**DOI:** 10.3390/molecules27185779

**Published:** 2022-09-07

**Authors:** João L. Serrano, Ana Maia, Adriana O. Santos, Eurico Lima, Lucinda V. Reis, Maria J. Nunes, Renato E. F. Boto, Samuel Silvestre, Paulo Almeida

**Affiliations:** 1CICS-UBI-Health Sciences Research Center, University of Beira Interior, Av. Infante D. Henrique, 6201-506 Covilhã, Portugal; 2Department of Chemistry, University of Beira Interior, Rua Marquês de Ávila e Bolama, 6201-001 Covilhã, Portugal; 3CQVR-Chemistry Centre of Vila Real, University of Trás-os-Montes and Alto Douro, Quinta de Prados, 5001-801 Vila Real, Portugal; 4CNC-Center for Neuroscience and Cell Biology, University of Coimbra, Rua Larga, 3004-517 Coimbra, Portugal

**Keywords:** cyanine dyes, colorectal cancer, antiproliferative agents, Caco-2 cells, cell cycle arrest

## Abstract

Cancer remains one of the diseases with the highest worldwide incidence. Several cytotoxic approaches have been used over the years to overcome this public health threat, such as chemotherapy, radiotherapy, and photodynamic therapy (PDT). Cyanine dyes are a class of compounds that have been extensively studied as PDT sensitisers; nevertheless, their antiproliferative potential in the absence of a light source has been scarcely explored. Herein, the synthesis of eighteen symmetric mono-, tri-, and heptamethine cyanine dyes and their evaluation as potential anticancer agents is described. The influences of the heterocyclic nature, counterion, and methine chain length on the antiproliferative effects and selectivities were analysed, and relevant structure–activity relationship data were gathered. The impact of light on the cytotoxic activity of the most promising dye was also assessed and discussed. Most of the monomethine and trimethine cyanine dyes under study demonstrated a high antiproliferative effect on human tumour cell lines of colorectal (Caco-2), breast (MCF-7), and prostate (PC-3) cancer at the initial screening (10 µM). However, concentration–viability curves showed higher potency and selectivity for the Caco-2 cell line. A monomethine cyanine dye derived from benzoxazole was the most promising compound (IC_50_ for Caco-2 = 0.67 µM and a selectivity index of 20.9 for Caco-2 versus normal human dermal fibroblasts (NHDF)) and led to Caco-2 cell cycle arrest at the G0/G1 phase. Complementary in silico studies predicted good intestinal absorption and oral bioavailability for this cyanine dye.

## 1. Introduction

Cancer is one of the top three causes of death in more than half of the world. About 19.3 million new cancer cases and 10.0 million cancer deaths were reported in 2020 in the last statistical study produced by the International Agency for Research on Cancer [1]. Within this data, breast, colorectal, and prostate cancers are among the top five types of cancers, and colorectal cancer is the second most deadly, only exceeded by lung cancer [1]. Beyond lifestyle and environmental factors, a hereditary genetic mutation can also be responsible for cancer diseases [1,2]. Several treatments have been used recently to overcome this public health issue, such as chemotherapy, radiotherapy, and photodynamic therapy (PDT) [2].

PDT is a relatively recent cancer therapy method that presents multiple advantages, especially in superficial tumours, compared with chemotherapy or radiotherapy [3,4]. This therapeutic approach is considered to be very effective, selective, non-invasive, and safe since the damage to normal tissues is minimal [3,4,5,6]. PDT employs light with a specific wavelength in the so-called “phototherapeutic window” (650–850 nm, allowing for optimal penetration of radiation into tissues), which, when combined with a photosensitiser and in the presence of molecular oxygen, produces cytotoxic reactive oxygen species [5,6,7,8] that induce selective malignant tissue destruction. Within the class of dyes generally considered for this purpose, cyanines are some of the most common near-infrared dyes evaluated as photosensitiser candidates for PDT [5,8,9].

Cyanine dyes, which are designated as such due to the beautiful blue colour (kyano means blue in Greek) of the first dye obtained accidentally by Williams Greville in 1856 [10], have two heterocyclic rings with at least one atom of nitrogen linked by a conjugation chain with an odd number of carbons. Their structure presents displaced electrons in one of the nitrogen atoms, which gives them a positive charge [11,12,13]. These dyes behave as resonance hybrids, giving them typical intense and narrow absorption spectral bands from the ultraviolet to infrared regions, covering the entire visible region. Cyanine dyes have applications in various science fields, mainly as functional dyes [14] that are used in high-technology applications, in opposition to well-known traditional applications as in the textile area, where the colour itself is the important application property. Representative examples are found in areas such as biochemistry, biotechnology, biology, physics, pharmacology, and medicine [11,13,15]. Some general reviews regarding their synthesis, properties, and functional applications can be found in the literature [8,11,12,13,16,17].

During the last decade, our research group has devoted some of its research work to cyanine dyes, including their synthesis [18,19,20,21,22]; structure analysis [18]; and functional application as hybrid materials [23], affinity chromatography ligands [24], probes for proteins detection [25], systems for sensitive pH determination [26], and as photodynamic agents in cancer treatment [20,21,22].

Although several studies on cyanine dyes’ potential applications for PDT were published [8,9,16,17], only a very restricted set showed a possible antiproliferative effect by itself, viz., in the absence of light and/or out of the “phototherapeutic window” [27,28,29,30,31]. Within the few examples found in the literature and to the best of our knowledge, Deligeorgiev and co-workers presented some promising antiproliferative activity effects of monomethine cyanine dyes against several cancer cell lines [29,30,31]. Benzothiazole, benzoxazole, and benzothiazolopyrimidine moieties were associated with the highest antiproliferative effects, where half-maximal inhibitory concentration (IC_50_) values of 0.06 and 0.001 µM for solid tumour and leukemia cell lines were determined, respectively. Additionally, some of these dyes presented high selectivity for cancer cells compared with normal ones [30]. On the other hand, monomethine cyanines stand out as non-toxic fluorescent dyes for nucleolar RNA [27] and mitochondria [28] imaging in living cells. 

Considering this window of opportunity, herein we report the synthesis, structural characterisation, and antiproliferative activity evaluation of a set of representative symmetrical mono-, tri-, and heptamethine cyanine dyes, as well as an investigation of the structure–activity relationships (SARs) regarding the nature of the heterocycles, counterions, and methine lengths concerning their potency and selectivity. Furthermore, the influence of light on the antiproliferative activity of the most promising dye, as well as its effect on cell morphology, apoptosis induction, and cell cycle arrest, were also assessed and discussed.

## 2. Results and Discussion

### 2.1. Synthesis and Structural Characterisation

Bisbenzothiazole **1** and mono- **2**–**14**, tri- **15** and **16**, and heptamethine **17**–**19** cyanine dyes containing benzothiazole, benzoxazole, benzoselenazole, indole, and quinoline as terminal nitrogen heterocycles, were successfully synthesised using or based on classical methods (Figure 1, Figure 2 and Figure 3) [32,33,34,35,36,37,38]. To the best of our knowledge, monomethine cyanines **8**–**11** and **14** were never described. Furthermore, the melting point (m.p.) values of **5**, **6**, **17**, and **18**; proton nuclear magnetic resonance (^1^H NMR) of **2**–**5**, **7**, **13**, and **15**–**18**; and carbon nuclear magnetic resonance (^13^C NMR) characterisation of **2**–**7**, **13**, and **15**–**18** are also described for the first time. Additionally, synthesis and full characterisation of the dye **12** in the form of tosylate salt (TsO^–^) is also presented as a novelty herein.

The synthesis of bisbenzothiazole **1** involved the condensation of 2-aminothiophenol with diethyl malonate (Figure 1) [32]. Although this compound can be used as a precursor for synthesising monomethine cyanine dyes [39], it was synthesised to be included in the SAR studies related to cyanine dyes’ biological evaluations.

A series of symmetrical monomethine cyanine dyes **2**–**14** were synthesised in three different ways. The process most frequently described in the literature is the thioalkyl method, which was used to prepare the dyes **2**–**7**, **12**, and **13**. In this one-pot synthesis method, the dyes **2**–**5** were prepared from the quaternisation of both 2-methylbenzothiazole and 2-methylthiobenzothiazole via melting at 120 °C in a mixture with methyl tosylate [33], followed by their condensation in the presence of pyridine. The dyes **2**–**5** were finally isolated and loaded with different counteranions (W = Cl, Br, I, ClO_4_, TsO; Figure 2 and Figure 3) in good to excellent yields (68 to 94%, Table 1). The dye **12** was also prepared via the same method using the related benzoxazole, but produced a lower yield (66%; Table 1). The *N*-ethyl cyanine dyes **6** and **7** were synthesised from 2-methylbenzothiazole and ethyl tosylate in the presence of 2-ethylthiobenzothiazole as an alternative to their congener 2-methylthiobenzothiazole. Nevertheless, the same reactivity was expected: a crossed alkylation was observed during the quaternary ammonium salts formation, affording the two symmetrical dyes **5** and **6** in a mixture with the corresponding asymmetric one [33]. 2,2-Quinolinium dye **13** was produced using the same thioalkyl method starting from 2-methylquinoline and 2-methylthioquinolinium, with a good yield of 76% (Table 1). On the other hand, 4,4-quinolinium dye **14** was prepared by the alternative potassium hydroxide method [34], with a satisfactory yield of 58% (Table 1). This method involved the ethanolic aqueous potassium hydroxide reflux of a mixture of isolated quinolinium and 4-methylquinolinium quaternary ammonium salts (Figure 3). Finally, dyes **8**–**11** were synthesised using the nitrite method [35] in low to good yields (25 to 75%, Table 1). This method allows the formation of symmetric monomethine cyanine dyes, sparing the presence of the alkylthio leaving group in the nitrogen neighbour that is required in the thioalkyl method. Despite the low reactional yields observed, the simplicity of this method is an asset in synthesising symmetric monomethine cyanine dyes, presenting an alternative to the traditional thioalkyl method.

To infer the influence of the conjugated chain size in the SAR study, trimethine **15** and **16** and heptamethine cyanine dyes **17**–**19** were also prepared (Figure 2). The first ones were synthesised using the orthoformate method through the condensation reaction between the corresponding 2-methylbenzoazolium salts and triethyl orthoformate in dry refluxing pyridine [36,37]. Heptamethine cyanine dyes **17**–**19** were prepared via an adaptation of the bisaldehyde method, where molecular sieves were used to remove the water that formed [38] as an alternative use of a Dean–Stark apparatus [43]. Thus, these dyes were prepared via the condensation reaction between the respective 2-methylbenzoazolium salts and 2-chloro-1-formyl-3-(hydroxymethylene)cyclohex-1-ene in the mixture of butan-1-ol/benzene (7:3) at reflux in the presence of dry pyridine and activated 4 Å molecular sieves.

The ^1^H and ^13^C NMR spectroscopic characterisations of bisbenzothiazole **1** and all the cyanine dyes **2**–**19** were in accordance with the expected structures and were supported by high-resolution mass spectrometry (HRMS) for the new compounds. Characteristic mesomethine and N^1/2+^CH_n_ ^1^H and ^13^C NMR signals are presented in Appendix A. The ^1^H NMR spectra evidence the nature of the methine dyes via the presence of a singlet between 5.80 and 7.24 ppm for the cyanine (monomethine) dyes **2**–**14** via the doublet and triplet at 6.09 to 6.79 and 7.81 to 8.32 ppm, respectively, for the carbocyanine (trimethine) dyes **5**–**16**, and via the two doublets in the ranges from 6.37 to 6.60 and 7.70 to 8.26 ppm for the tricarbocyanine (heptamethine) dyes **17**–**19**. The typical unshielded proton and carbon linked to the ammonium atom appeared at 4.02–5.91 and 39.0–49.1, respectively, according to the nature of the terminal nitrogen heterocycles. 

As expected [44], the wavelengths of maximum absorption λ_max_ (nm) in the ultraviolet-visible (UV-vis) spectra remained practically unchanged when the nature of the counteranion or the *N*-substituent groups varied (Table 1). On the other hand, and also expected [44], a bathochromic shift around 100 nm was observed for each additional vinyl group in the methine chain (λ_max_: cyanines **2**–**12** from 376 to 427 nm, carbocyanines **15** and **16** from 485 to 573 nm, and tricarbocyanine 1**7**–**19** from 787 to 814 nm). 

Considering that all cyanine dyes **2**–**16** presented absorption outside the therapeutic window, the antiproliferative activity in the absence of light of all dyes was evaluated. In addition, the influence of light as a potential photodynamic synergistic effect on the antiproliferative activity was also assessed and is discussed below.

### 2.2. In Vitro Studies

#### 2.2.1. Antiproliferative Effects in Human Cell Lines

As a preliminary study regarding the antiproliferative potential of the precursor **1** and the cyanine dyes **2**–**19** while envisioning their potential interest in cancer treatment, a 3-(4,5-dimethylthiazol-2-yl)-2,5-diphenyltetrazolium bromide (MTT) assay was performed in human tumour cell lines of colorectal (Caco-2), breast (MCF-7), and prostate (PC-3) adenocarcinomas and a non-tumourous cell line (normal human dermal fibroblasts (NHDF)). This evaluation was performed at a single concentration of 10 µM, and the anticancer drug 5-fluorouracil (5-FU) was used as a positive control.

In an analysis of the screening results (Figure 1 and Appendix A), the structural influence of studied dyes on the antiproliferative behaviour was clearly observed for the cell lines in the study. In general, all cyanine dyes decreased the percentage of cell viability in the cancer cell lines, which can be partially explained by their lipophilic structure with a delocalised positive charge between their nitrogen atoms, which allowed them to permeabilise the lipid bilayers [8,45]. In addition, the cationic nature of cyanine dyes easily enables them to penetrate the negative mitochondrial membrane and interfere with the membrane potential, causing a disruption in the functioning of this organelle, leading to a breakdown of energy production and the death of the cell [8,46]. Moreover, the cytotoxicity associated with cancer cells’ mitochondria can lead to some selectivity over normal cells due to the well-known characteristic of cyanine dyes to accumulate and be retained in tumours. This is especially relevant in cases such as PDT treatment, where photosensitisers are non-toxic, and cells can be exposed to these compounds for a long time before the application of irradiation [8,47,48], which was not the case in this study.

From a more detailed analysis of Figure 1 and Appendix A, some SAR data could be inferred. The antiproliferative potential of the monomethine benzothiazole cyanine dyes **2**–**11** compared with bisbenzothiazole **1** reinforced the idea that the structural nature of the cyanine dyes, namely, the positive charge, as well as the presence of the *N*-substituent groups, were decisive for the prominent cytotoxicity of cyanine dyes. Monomethine dyes **2**–**8**, **10**, and **11** strongly inhibited cellular growth at 10 µM, while bisbenzothiazole **1** only inhibited it at nearly 50%. The unique exception was the monomethine dye **9**, which showed an effect on cellular proliferation equivalent to bisbenzothiazole **1**, revealing that the presence of an ester as the *N*-substituent group was counterproductive for antiproliferative activity. In this first screening assay, the benzothiazole **2**–**8**, **10**, **11**, and **17** and benzoselenazole **16** cyanine dyes seemed to be more active than their congener benzoxazole **12** and **15** and quinolinium **13** and **14** dyes**.** Nonetheless, the benzoxazole monomethine dye **12** appeared to be particularly selective for cancerous PC-3, MCF-7, and Caco-2 cell lines relative to NHDF. In general, the heptamethine dyes **18** and **19** were less cytotoxic than the monomethine derivatives, while the heptamethine **17** and trimethine dyes **15** and **16** presented an equivalent effect on cellular viability. Finally, apart from bisbenzothiazole **1** and the dyes **9**, **18**, and **19**, all the dyes presented higher cytotoxicity than the well-known anticancer drug 5-FU.

Considering this initial screening where the monomethine dye **12** was revealed to be especially selective for cancer over non-tumoural cells, although not the most cytotoxic within the dyes of this set, this dye was selected for further studies to determine the IC_50_ in all four cell lines under investigation. The determined IC_50_ values (horizontal grey line in Table 2, and Figure 2) for this dye were 4.53, 4.57, and 0.67 µM for the PC-3, MCF-7, and Caco-2 cancer cell lines, respectively, and 13.95 µM for the NHDF cell line, corroborating the selectivity pointed out by the initial screening results. 

The benzoxazole dyes **15** and, remarkably, **12** presented a high selectivity index (SI) for all cancer cell lines under study (**15:** SI Caco-2 10.4, **12**: SI Caco-2 20.9, SI MCF-7 4.5, SI PC-3 4.53). The remarkable SI value of 20.9 observed for **12** encouraged us to evaluate the antiproliferative effects on the colon Caco-2 cells of all the dyes (**2**–**8** and **10**–**17**) that led to cell viability lower than 30% in the initial 10 µM screening in this cell line. In Table 2, the calculated IC_50_ values for the dyes **2**–**8** and **10**–**17** and 5-FU as a positive control for normal NHDF and cancer Caco-2 cell lines are presented, as well as the SI for Caco-2 versus NHDF cell lines; some noteworthy and promising conclusions can be drawn, which are outlined below.

Almost all dyes under study were significantly more cytotoxic than the 5-FU positive control. In addition, this study confirmed the selectivity observed for the dye **12** in the screening. The highest SI was found for this derivative, which was clearly higher than that calculated for the other cyanine dyes. Interestingly, SI values of 10.4 and 6.5 were found for the benzoxazole and benzoselenazole cyanine dyes **15** and **16**, respectively. According to our experience, the nature of the counterions affects the solubility of cyanine dyes to some extent but does not significantly affect their chemical properties. However, considering the data in Table 2, the nature of the counterion seemed to influence the cell viability in both non-tumoural and cancerous cells, which was in line with what was previously described in the literature [49]. In fact, in a more detailed analysis of Caco-2 cell line viability, different cytotoxicity could be observed for the tosylate dye **5** (IC_50_ = 0.28 µM), the iodide dye **4** (IC_50_ = 0.40 µM), the bromide dye **3** (IC_50_ = 0.52 µM), and the chloride dye **2** (IC_50_ = 0.65 µM), suggesting that the size of the anion could have been of relevance. This result seemed to be in agreement with previous studies with disinfectants containing these ions in mammalian cells [50], as well as for 1-butyl-3-methylimidazolium chloride, -bromide, -iodide, and -tosylate ionic liquids on a promyelocytic leukemia rat cell line [51]. In the NHDF cell line, the effects on cell viability also depended on, to some extent, the size of the counterion, with the tosylate dye **5** being more cytotoxic than the chloride dye **2**. In opposition, the iodide dye **4** presented lower cytotoxicity against the NHDF cells than the bromide dye **3**, contrary to what happens against Caco-2 cells. Concerning the effect of the counterion on the selectivity, both perchlorate and tosylate appeared to be the salts with higher SI between the cancerous Caco-2 and non-cancerous NHDF cell lines.

The natures of the *N*-substituents also seemed to have some influence on both the cytotoxicity and SI of the cancerous Caco-2 and non-cancerous NHDF cell lines. Along the iodide benzothiazole cyanine **4** (IC_50_ = 0.40 µM, SI = 1.8), **8** (IC_50_ = 0.95 µM, SI = 1.0), **9** (IC_50_ = 1.00 µM, SI = 0.7), and **10** (IC_50_ = 1.24 µM, SI = 0.6) series, from the longest chain to the methyl group, it was clear that the decrease in the length of the *N*-substituents led to an increase in both the cytotoxicity and the SI.

In addition to the dependence on the nature of the counterion and the *N*-substituents of the cyanine dyes regarding the antiproliferative effects, the influence of the nature of the heterocyclic bases strongly determined both the cytotoxicity and selectivity of the cell lines. In fact, along the series of the 2,2′-quinolinium **13** (IC_50_ = 0.58 µM, SI = 1.0), benzothiazolium **4** (IC_50_ = 0.40 µM, SI = 1.8), benzoxazolium **12** (IC_50_ = 0.67 µM, SI = 20.9), and benzothiazolium **5** (IC_50_ = 0.28 µM, SI = 2.4) dyes, while the benzothiazolium dyes tended to be more cytotoxic, the benzoxazolium ones were more selective. Additionally, the benzoxazolium carbocyanine dye **15** (IC_50_ = 0.39 µM, SI = 10.4) was less potent but more selective than its counterpart benzoselenazolium **16** (IC_50_ = 0.07 µM, SI = 6.5). Therefore, the use of the benzoxazolium moiety as an alternative for the quinolinium, benzothia-, or benzoselenazolium nucleus was clearly the best choice in terms of cytotoxicity and selectivity.

Analysing the results displayed in Table 2 for the monomethinecyanines **2**–**11** and **13**, carbocyanines **15** and **16**, and the tricarbocyanine **17**, the nature of the methinic chain by itself did not seem to present a decisive influence on both the cytotoxicity and SI. Nevertheless, when comparing tricarbocyanine **17** (IC_50_ = 2.06 µM, SI = 0.4) to monomethinecyanine **10** (IC_50_ = 1.00 µM, SI = 0.7), a slight increase in cytotoxicity and SI was found with the reduction in the methinic chain.

Although the trimethine cyanine dyes **15** and **16** were found to be more potent in the Caco-2 cells than the monomethine cyanine dye **12**, the latter showed the best compromise between potency against the Caco-2 cell line and selectivity for cancer against non-cancer cells. In conclusion, and looking in a global sense at all the results presented above (Table 2), the monomethine cyanine dye **12** was here appointed as the most promising compound in this study and was selected for further studies.

#### 2.2.2. Photocytotoxicity and Photostability Evaluation

As the use of cyanine dyes as anti-cancer agents is mainly associated with PDT [8,9], and despite monomethine and almost all the more simple tricarbocyanine dyes being outside of the so-called “phototherapeutic window” (650–850 nm) [5,6,7,8], we also evaluated the photocytotoxicity of the most promising monomethine cyanine dye **12**. This evaluation was performed in the NHDF cell line using the MTT method after a light incidence to evaluate an eventual synergistic contribution to the cytotoxicity of this dye.

For the light source, a commercially available 30 W RGB LED system with white light irradiation was used. Two controls were performed to validate the potential applicability of this newly built system. First, the emission spectrum of this light source was analysed to guarantee the emission at the maximum absorbance wavelength of the dye under study. The spectra of all four lights (white, red, green, and blue) were measured and are presented in Appendix A. Second, the photocytotoxicity against NHDF cells of a squaraine cyanine dye that was previously reported by our research group as a potential antitumoural phototherapeutic agent [21], but now using other available light sources, was performed as a positive control. The results (Appendix A) showed clear differences between assays in the dark versus after 30 min of irradiation, as previously described [21]. For the positive control (herein called EL4), an IC_50_ close to the prior cited value was determined, with a value of 4.16 µM versus the described 3.28 µM [21]. These results showed that this new light source was suitable for evaluating the phototoxicity of the monomethine cyanine dye **12** in a non-cancerous NHDF cell line (Figure 3).

The analysis of irradiation influence on the cytotoxicity of the dye **12** (Figure 3) demonstrated a total absence of photocytotoxicity. In opposition to what was observed for the positive control EL4 (Appendix A), where an increase in cytotoxicity was quite obvious in the light’s presence, the dye **12** even showed a slight cytotoxicity decrease. This result suggested a possible photodegradation of the dye **12** in the presence of light, which was confirmed via a photostability assay that was performed in phosphate-buffered saline (PBS) to mimic the physiological environment.

Indeed, the cyanine dye **12** suffered slight decomposition (approximately 4 and 5% after 30 and 120 min of irradiation, respectively), as demonstrated by the decrease in absorbance at its maximum absorption wavelength in the presence of a light source over 120 min (Figure 4). Therefore, this slight decomposition of the dye **12** could explain the slight increase in NHDF proliferation caused by the presence of light for 30 min. However, the expected absence of light during the eventual use of monomethine cyanine dyes for treatment of internal cancer cells tissues may prevent the residual photodegradation of these dyes. Moreover, the absence of any photodynamic action prevents the most common side effect of PDT, namely, the sensitivity to bright lights and sunlight, which is an additional advantage to the potential use of this family of cyanine dyes as anti-proliferative agents.

#### 2.2.3. Morphological Analysis of Caco-2 Cells

It is well known that cell morphological changes may point to the potential of an associated cell death mechanism [52,53]. Therefore, microscopy images of Caco-2 cells revealed that exposure to 10 µM of 5-FU and 1 or 10 µM of the dye **12** after 24, 48, and 72 h (Figure 5 and Figure 6) led to morphological changes compared with the negative control cells. At the microscopic magnification of 10× (Figure 5), a cell population increase over time of the Caco-2 untreated cells was visible. On the other hand, the decrease in these same populations was noticeable when the cells were treated with 10 µM of 5-FU or 1 and 10 µM of the dye **12**. Some shrinking cells and few apoptotic bodies after 72 h of treatment when these cells were exposed to 5-FU at 10 µM were also visible. When observing the cellular morphology with higher magnifications of 20 or 40× after 72 h of incubation (Figure 6), a cell size increase with an enlarged and well-defined nucleus was also noted. Moreover, some differences were noticed when the dye **12** was used compared with untreated cells. Although there were no visible differences at the concentration of 1 µM (Figure 6), at 10 µM, the dye **12** induced an evident contraction, and thus, a decrease in cell size. Additionally, some shrinking cells and apoptotic bodies were visible after the treatment with 10 µM of the dye **12** (Figure 5), although in a much lower number than those observed for the 5-FU, but more noticeably than in the untreated cells.

#### 2.2.4. Flow Cytometry Studies of Apoptosis and Cell Cycle Effects

Considering apoptosis as a common programmed mechanism of cell death and that apoptotic cells are characterised by the loss of nuclear content caused by DNA fragmentation, we used the method described by Riccardi and Nicolleti [54] to analyse the eventual apoptosis induced by the dye **12** via flow cytometry with propidium iodide staining. According to our previous results, a concentration of 10 µM of the dye **12** was used since this concentration was expected to cause meaningful cellular morphological changes. Additionally, 600 mM of sorbitol was used as an apoptosis-positive control, and 10 µM of 5-FU was used to compare with the dye **12.** The dye **12** induced apoptosis in about 3% of cells (Figure 7A). However, although statistical significance was determined relative to the negative control (untreated cells), it represented only a slight increase of 2% from the verified residual apoptosis. On the other hand, the positive control sorbitol was found to induce an extent of apoptosis of about 28%, in agreement with what was previously described for other human colorectal cancer cell lines [55]. Furthermore, as expected and as previously published in the literature [56], incubation with 5-FU also induced an increase in apoptotic cells by approximately 10%. Therefore, apoptosis should not be considered the preferred mechanism of the high cytotoxic activity observed for the dye **12**.

After the exclusion of subG1 events, we also evaluated the influence of all three compounds on the cell cycle distribution. In this assay, 10 µM of 5-FU was used as a positive control for S-phase arrest [56]. As shown in Figure 7B, 600 mM of sorbitol did not significantly affect the Caco-2 cell cycle. On the other hand, 5-FU at 10 µM caused a significant increase in the rate of Caco-2 cells in the S-phase and a consequent decrease in the percentage of cells in the G0/G1 phase with no relevant effect on the G2/M phase. Finally, treatment with the dye **12** induced a notorious increase in the number of cells in the G0/G1 phase and a decrease in the S and G2/M phases compared with untreated Caco-2 cells. These results agreed with the morphological changes that were observed and discussed above. In opposition to what was evidenced in the 5-FU assay, where a morphological cellular enlargement and well-defined nucleus were observed, which was possibly caused by DNA synthesis and a consequent cellular increase in the S-phase, the contraction and decrease in cell size promoted by the dye **12** could be explained by the observed G0/G1 phase arrest.

### 2.3. Computational Studies

Poor pharmacokinetic profiles or low efficacy are the major failures in developing drug candidates into drugs [57]. To complement the in vitro biological results, an in silico study for the dye **12** was performed to predict the physicochemical and druglikeness properties. The online tool SwissADME [58] was used to analyse simple rules and filters based on molecular properties and predict the future in vivo applicability of the dye **12**. The first one, namely, Lipinski’s rule, which is known as the “rule of five”, relates the potential pharmacokinetics with several physicochemical properties of molecules. Lipinski showed that compounds with more than 5 hydrogen bond donors (HBdonor), 10 hydrogen bond acceptors (HBacceptor), a molecular weight (MW) greater than 500 g/mol, and the calculated logarithm of 1-octanol/water partition coefficient (CLogP) greater than 5 showed potential poor absorption or permeation [59]. After this approach, several variants were described to improve the druglikeness predictions. In this way, Ghose’s rule defines an organic compound with MW between 160 and 480 g/mol, log P between −0.4 and 5.6, molar refractivity between 40 and 130, and a total number of atoms between 20 and 70 can be a druglike molecule [60]. On the other hand, Veber’s rule positively correlates a high probability of a good oral bioavailability in rats for compounds that meet the criteria of 10 or fewer rotatable bonds and a topological polar surface area (TPSA) equal to or less than 140 Å^2^ [61]. As stated in Egan’s rule, a high level of human intestinal absorption was also correlated with compounds that have CLogP less than 5.88 and a TPSA less than 131.6 Å^2^ [62]. Finally, Muegge’s rule applies a simple pharmacophore point filter to discriminate druglike from non-druglike compounds. This filter is based on simple structural rules, and several well-defined pharmacophore points between two and seven are required to pass the filter [63]. Considering all these rules, Table 3 condensed all the results determined for the dye **12** to demonstrate its fulfilment for all the abovementioned druglikeness filters and, therefore, its potential as a druglike compound that has good intestinal absorption and good oral bioavailability.

To predict the potential instability, high reactivity, and promiscuity of the dye **12** for several biological targets, which can be associated with a nonspecific mode of action, we also applied some medicinal chemistry filters using the SwissADME tool for the identification of potentially problematic fragments. The results (Table 3) also showed that the dye **12** should not belong to the set of pan assay interference compounds (PAINS) [64]. However, an alert was found using the Brenk filter [65] due to quaternary nitrogen fragments, which are occasionally associated with some toxicity problems [66]. Although this positive charge is just partial since it is shared between the two nitrogen atoms and is delocalised over the polymethine chain, this alert must be considered in future biological studies.

## 3. Materials and Methods

### 3.1. Chemistry

All reagents and solvents were purchased from commercial suppliers and used without further purification. 2-Chloro-1-formyl-3-(hydroxymethylene)cyclohex-1-ene was prepared according to the procedure in [67] with minor modifications, while benzyl 4-methylbenzenesulfonate was synthesised from benzyl alcohol and 4-toluenesulfonyl chloride in pyridine [68]. Isoamyl nitrite was freshly prepared from isoamyl alcohol and sodium nitrite [69]. 1-Alkyl-2-methylthiobenzoazolium, 1-alkyl-2-methylthioquinolinium, 1-alkyl-2-methylbenzoazolium, and 1-alkyl-2-methylquinolinium salts were prepared according to the previously described method [70,71]. Pyridine, butan-1-ol, and benzene were dried before use via standard methods [72] and kept over molecular sieves (4 Å, beads, 4–8 mesh). All reactions were monitored via TLC using Macherey–Nagel 60 G/UV_254_ (0.2 mm) plates, which were eluted in dichloromethane or dichloromethane/methanol (9:1) and were visualised under UV radiation at 254 nm. The m.p. values were measured on a Büchi B-540 apparatus and were uncorrected. UV-vis spectra were recorded for all dyes under study on a Thermo Scientific Evolution 160 UV-VIS spectrophotometer using ethanol as a solvent. The wavelength of maximum absorption was reported in nm. The infrared (IR) spectra were obtained with a Thermo Fisher Scientific Nicolet iS10: smart iTR infrared spectrophotometer using KBr pellets and were processed with OMNIC 8.2 software. NMR spectra were acquired on a Brüker Avance III 400 MHz spectrometer (^1^H NMR at 400.13 MHz and ^13^C NMR at 100.62 MHz) and were processed with the software MestReNova 11.0.3 (trial). The chemical shift (δ) values are given in parts per million (ppm) and coupling constants (*J*) in hertz (Hz). The multiplicity of the signals is reported as singlet (s), doublet (d), doublet of doublets (dd), doublet of triplets (dt), doublet of doublet of doublets (ddd), triplet (t), triplet of doublets (td), quartet (q), pentet (p), sextet (sx), or multiplet (m). Dimethyl sulfoxide hexadeutered (DMSO-*d*_6_) was used as a solvent and internal standard (2.50 and 39.52 ppm in ^1^H and ^13^C NMR, respectively). HRMS was performed for new compounds via electrospray ionisation time-of-flight (ESI-TOF) using the NUCLEUS services at the University of Salamanca (Spain).

#### 3.1.1. Synthesis of bis(benzo[d]thiazol-2-yl)methane (**1**) 

A mixture of 2-aminothiophenol (214 µL; 250 mg, 2 mmol) and diethyl malonate (153 µL; 160 mg; 1 mmol) were heated at 170 °C for 5 h and 30 min. After cooling, the resulting solid was crystallised from ethanol to obtain the desired product [32]. Yield: 73%; yellow crystals; m.p. 97–98 °C (lit. 95–96 °C [32]); IR λ_max_ (KBr): 3060, 2988, 1678, 1490, 1335, 1095, 760, 730 cm^−1^; ^1^H NMR (400 MHz, DMSO-*d*_6_): δ 8.09 (d, *J* = 7.9 Hz, 2H), 8.00 (d, *J* = 8.0 Hz, 2H), 7.52 (ddd, *J* = 8.3, 7.2, 1.3 Hz, 2H), 7.45 (ddd, *J* = 8.3, 7.2, 1.3 Hz, 2H), 5.10 (s, 2H); ^13^C NMR (101 MHz, DMSO-*d*_6_): δ 166.3, 152.6, 135.2, 126.3, 125.3, 122.6, 122.3, 37.8.

#### 3.1.2. General Procedures (GP) for the Synthesis of Monomethine Cyanine Dyes **2**–**14**

GP1—thioalkyl method via one-pot synthesis [33]: a mixture of 2-methylbenzoazole (1.0 mmol), 2-methyl or 2-ethylthiobenzoazole (1.0 mmol), and methyl or ethyl tosylate (2.2–3.5 mmol) was heated at 120–140 °C for 1–6 h until an almost total conversion into the respective quaternary ammonium salts was achieved. The resulting mixture of solid salts was refluxed in dry pyridine (10 mL of pyridine/1 g of salt). After cooling to room temperature, diethyl ether was added to allow for complete precipitation. The resulting crystalline product was collected via filtration under reduced pressure, washed with diethyl ether (3 × 3 mL), and recrystallised from methanol or ethanol.

GP2—nitrite method [35]: to a mixture of the corresponding 2-methylbenzothiazolium salt (1 mmol) in boiling acetic anhydride (2 mL), freshly prepared isoamyl nitrite (120 µL) was added, which produced a violent formation of foam while the quaternary salt was dissolved and afforded a dark brown solution. The mixture was maintained under constant stirring until room temperature was reached. The formed solid was filtered under reduced pressure and washed with water (3 mL) and diethyl ether (3 × 3 mL). The obtained solid was dried and recrystallised from methanol or ethanol.

GP3—thioalkyl method [32]: a mixture of a 1-alkyl-2-methylthiobenzoazolium or 1-alkyl-2-methylthioquinolinium salt (1 mmol), and 1-alkyl-2-methylbenzoazolium or 1-alkyl-2-methylquinolinium salt (1 mmol) in pyridine (10 mL of pyridine/g of salt) or ethanol (2 mL) and triethylamine (2.2 mmol) was refluxed for 1–5 h. After the reaction mixture had cooled, the formed solid was filtered under vacuum, washed with ethyl ether (3 × 3 mL), and dried. After that, diethyl ether was added to the suspension to allow for the complete precipitation of the product so formed. The obtained solid was dried and recrystallised from methanol or ethanol.

GP4—potassium hydroxide method [34]: the solution of 2- or 4-methylquinolinium salt (1 mmol), quinolinium salt (3 mmol), and potassium hydroxide (aqueous solution, 3 mmol, 168 mg) in ethanol (6 mL) was refluxed for 15 min. After cooling, the resulting solid was filtered under reduced pressure and recrystallised from methanol or ethanol.


*Methyl-2-((3-methylbenzo[d]thiazol-2(3H)-ylidene)methyl)benzo[d]thiazol-3-ium chloride* (**2**): using GP1, from 2-(methylthio)benzothiazole (181 mg), 2-methylbenzothiazole (149 mg), and methyl tosylate (409 mg). Anion exchange was caused by reflux in a hydrochloric acid solution 18.5% (10 mL) or a saturated sodium chloride solution (10 mL). Yield: 68%; yellow crystals; m.p. 265–266 °C (lit. 269 °C [32]); vis λ_max_ (EtOH): 424 nm, log ε = 4.97; IR λ_max_ (KBr): 3396, 1532, 1470, 1358, 1281, 756 cm^−1^; ^1^H NMR (400 MHz, DMSO-*d*_6_): δ 8.20 (dd, *J* = 8.0, 1.2 Hz, 2H), 7.86 (d, *J* = 8.3 Hz, 2H), 7.66 (ddd, *J* = 8.1, 7.4, 1.0 Hz, 2H), 7.48 (t, *J* = 7.6 Hz, 2H), 6.70 (s, 1H), 4.02 (s, 6H); ^13^C NMR (101 MHz, DMSO-*d*_6_): δ 162.1, 140.8, 128.6, 125.0, 124.8, 123.5, 113.9, 83.0, 34.2.*3-Methyl-2-((3-methylbenzo[d]thiazol-2(3H)-ylidene)methyl)benzo[d]thiazol-3-ium bromide* (**3**): using GP1, from 2-(methylthio)benzothiazole (181 mg), 2-methylbenzothiazole (149 mg), and methyl tosylate (409 mg). Anion exchange was caused by reflux in a saturated sodium bromide solution (10 mL). Yield: 69%; yellow crystals; m.p. 302–303 °C (lit. 292–293 [40]); vis λ_max_ (EtOH): 424 nm, log ε = 4.83; IR λ_max_ (KBr): 3396, 1524, 1470, 1354, 1281, 756 cm^−1^. ^1^H NMR (400 MHz, DMSO-*d*_6_): δ 8.22 (dd, *J* = 8.0, 1.1 Hz, 2H), 7.88 (d, *J* = 8.3 Hz, 2H), 7.68 (ddd, *J* = 8.1, 7.3, 1.1 Hz, 2H), 7.50 (t, *J* = 7.7 Hz, 2H), 6.72 (s, 1H), 4.03 (s, 6H); ^13^C NMR (101 MHz, DMSO-*d*_6_): δ 162.1, 140.8, 128.5, 124.9, 124.8, 123.5, 113.8, 82.9, 34.2.*3-Methyl-2-((3-methylbenzo[d]thiazol-2(3H)-ylidene)methyl)benzo[d]thiazol-3-ium iodide* (**4**): using GP1, from 2-(methylthio)benzothiazole (181 mg), 2-methylbenzothiazole (149 mg), and methyl tosylate (409 mg). Anion exchange was caused by reflux in a saturated potassium iodide solution (10 mL). Yield: 94%; yellow crystals; m.p. 315–316 °C (lit. 303 °C [33]); vis λ_max_ (EtOH): 424 nm, log ε = 4.96; IR λ_max_ (KBr): 3438, 1524, 1477, 1358, 1285, 758 cm^−1^. ^1^H NMR (400 MHz, DMSO-*d*_6_): δ 8.21 (dd, *J* = 8.0, 1.2 Hz, 2H), 7.87 (d, *J* = 8.3 Hz, 2H), 7.67 (ddd, *J* = 8.4, 7.3, 1.2 Hz, 2H), 7.49 (ddd, *J* = 8.2, 7.3, 1.0 Hz, 2H), 6.70 (s, 1H), 4.02 (s, 6H); ^13^C NMR (101 MHz, DMSO-*d*_6_): δ 162.1, 140.8, 128.5, 124.9, 124.8, 123.5, 113.9, 82.9, 34.1.*3-Methyl-2-((3-methylbenzo[d]thiazol-2(3H)-ylidene)methyl)benzo[d]thiazol-3-ium 4-methylbenzenesulfonate* (**5**): using GP1, from 2-(methylthio)benzothiazole (181 mg), 2-methylbenzothiazole (149 mg), and methyl tosylate (409 mg). Yield: 77%; yellow crystals; m.p. 296–297 °C; vis λ_max_ (EtOH): 424 nm, log ε = 4.98; IR λ_max_ (KBr): 3384, 1632, 1536, 1474, 1362, 1282, 752 cm^−1^; ^1^H NMR (400 MHz, DMSO-*d*_6_): δ 8.21 (dd, *J* = 8.1, 1.2 Hz, 2H), 7.88 (d, *J* = 8.3 Hz, 2H), 7.69 (ddd, *J* = 8.4, 7.3, 1.2 Hz, 2H), 7.50 (t, *J* = 7.3 Hz, 2H), 7.47 (d, *J* = 8.0 Hz, 2H), 7.10 (d, *J* = 8.0 Hz, 2H), 6.72 (s, 1H), 4.03 (s, 6H), 2.27 (s, 3H); ^13^C NMR (101 MHz, DMSO-*d*_6_): 162.1, 145.8, 140.8, 137.5, 128.5, 128.0, 125.5, 124.9, 124.8, 123.5, 113.8, 82.9, 34.1, 20.8.*3-Methyl-2-((3-ethylbenzo[d]thiazol-2(3H)-ylidene)methyl)benzo[d]thiazol-3-ium 4-methylbenzenesulfonate* (**6**): using GP1, from 2-(ethylthio)benzothiazole (195 mg), 2-methylbenzothiazole (149 mg), and ethyl tosylate (441 mg). Yield: 80%; yellow crystals; m.p. 194–196 °C; vis λ_max_ (EtOH): 425 nm, log ε = 4.84; IR λ_max_ (KBr): 3068, 2997, 2923, 1529, 1465, 1373, 1339, 1269, 1192, 1128, 1034, 1011, 732, 681, 566 cm^−1^; ^1^H NMR (400 MHz, DMSO-*d*_6_): δ 8.23 (dd, *J* = 8.0, 1.2 Hz, 2H), 7.90 (d, *J* = 8.3 Hz, 2H), 7.70 (dt, *J* = 7.7, 1.1 Hz, 2H), 7.51 (t, *J* = 7.6 Hz, 2H), 7.47 (d, *J* = 8.0 Hz, 2H), 7.10 (d, *J* = 7.9 Hz, 2H), 6.75 (s, 1H), 4.70 (q, *J* = 7.1 Hz, 4H), 2.28 (s, 3H), 1.38 (t, *J* = 7.1 Hz, 6H); ^13^C NMR (101 MHz, DMSO-*d*_6_): δ 161.3, 145.8, 139.7, 137.6, 128.6, 128.0, 125.5, 125.0, 124.9, 123.5, 113.6, 81.9, 41.6, 20.8, 12.3.*3-Ethyl-2-((3-ethylbenzo[d]thiazol-2(3H)-ylidene)methyl)benzo[d]thiazol-3-ium perchlorate* (**7**): using GP1, from 2-(ethylthio)benzothiazole (195 mg), 2-methylbenzothiazole (149 mg), and ethyl tosylate (441 mg). Anion exchange was caused by reflux in a saturated sodium perchlorate solution (10 mL). Yield: 83%; yellow crystals; m.p. 296–297 °C; vis λ_max_ (EtOH): 425 nm, log ε = 4.92; IR λ_max_ (KBr): 3093, 2976, 2933, 1592, 1529, 1466, 1375, 1337, 1315, 1266, 1234, 1081, 747, 622 cm^−1^; ^1^H NMR (400 MHz, DMSO-*d*_6_): δ 8.23 (dd, *J* = 8.1, 1.2 Hz, 2H), 7.91 (d, *J* = 8.3 Hz, 2H), 7.70 (ddd, *J* = 8.4, 7.2, 1.3 Hz, 2H), 7.51 (t, *J* = 7.7 Hz, 2H), 6.75 (s, 1H), 4.70 (q, *J* = 7.2 Hz, 4H), 1.38 (t, *J* = 7.0 Hz, 6H); ^13^C NMR (101 MHz, DMSO-*d*_6_): δ 161.4, 139.7, 128.6, 125.0, 124.9, 123.5, 113.6, 81.9, 41.6, 12.3.*3-Benzyl-2-((3-benzylbenzo[d]thiazol-2(3H)-ylidene)methyl)benzo[d]thiazol-3-ium iodide* (**8**): using GP2, from 3-benzyl-2-methylbenzo[*d*]thiazol-3-ium iodide (367 mg). Yield: 25%; yellow crystals; m.p. 249–250 °C; vis λ_max_ (EtOH): 427 nm, log ε = 4.87; IR λ_max_ (KBr): 3019, 1562, 1471, 1407, 1355, 1309, 1282, 1218, 1159, 1131, 1025, 936, 819, 754 cm^−1^; ^1^H NMR (400 MHz, DMSO-*d*_6_): δ 8.28 (dd, *J* = 8.0, 1.2 Hz, 2H), 7.87 (d, *J* = 8.3 Hz, 2H), 7.66 (ddd, *J* = 8.5, 7.3, 1.3 Hz, 2H), 7.52 (t, *J* = 7.6 Hz, 2H), 7.33–7.17 (m, 6H), 7.14 (dd, *J* = 8.0, 1.4 Hz, 4H), 6.96 (s, 1H), 5.91 (s, 4H); ^13^C NMR (101 MHz, DMSO-*d*_6_): δ 162.6, 140.4, 134.1, 129.0, 128.8, 127.9, 126.7, 125.3, 125.0, 123.8, 114.1, 83.4, 49.1; ESI-HRMS calcd for [M-I]^+^ C_29_H_23_N_2_S_2_^+^ 463.1297, found 463.1289.*3-(2-Methoxy-2-oxoethyl)-2-((3-(2-methoxy-2-oxoethyl)benzo[d]thiazol-2(3H)-ylidene)methyl)benzo[d]thiazol-3-ium bromide* (**9**): using GP2, from 3-(2-methoxy-2-oxoethyl)-2-methylbenzo[*d*]thiazol-3-ium bromide (302 mg). Yield: 28%; yellow crystals; m.p. 226–227 °C; vis λ_max_ (EtOH): 427 nm, log ε = 5.13; IR λ_max_ (KBr): 3373, 3075, 2990, 2851, 1737, 1501, 1360, 1290, 1009, 844, 759, 520 cm^−1^; ^1^H NMR (400 MHz, DMSO-*d*_6_): δ 8.26 (d, *J* = 8.0 Hz, 2H), 7.82 (d, *J* = 8.3 Hz, 2H), 7.68 (t, *J* = 7.8 Hz, 2H), 7.54 (t, *J* = 7.7 Hz, 2H), 6.75 (s, 1H), 5.71 (s, 4H), 3.76 (s, 6H); ^13^C NMR (101 MHz, DMSO-*d*_6_): δ 166.9, 163.4, 140.2, 128.9, 125.5, 124.6, 123.8, 113.8, 83.4, 53.0, 47.5; ESI-HRMS calcd for [M-Br]^+^ C_21_H_19_N_2_O_4_S_2_^+^ 427.0781, found 427.0770.*3-Pentyl-2-((3-pentylbenzo[d]thiazol-2(3H)-ylidene)methyl)benzo[d]thiazol-3-ium iodide* (**10**): using GP2, from 2-methyl-3-pentylbenzo[*d*]thiazol-3-ium iodide (347 mg). Yield: 75%; yellow crystals; m.p. 164–166 °C; vis λ_max_ (EtOH): 426 nm, log ε = 4.77; IR λ_max_ (KBr): 3063, 2950, 2923, 2858, 1519, 1504, 1463, 1371, 1344, 1264, 1192, 746, 501 cm^−1^; ^1^H NMR (400 MHz, DMSO-*d*_6_): δ 8.22 (d, *J* = 7.9 Hz, 2H), 7.91 (d, *J* = 8.4 Hz, 2H), 7.69 (t, *J* = 7.8 Hz, 2H), 7.51 (t, *J* = 7.6 Hz, 2H), 6.68 (s, 1H), 4.65 (t, *J* = 7.5 Hz, 4H), 1.79 (p, *J* = 7.6 Hz, 4H), 1.44 (p, *J* = 7.0 Hz, 4H), 1.35 (sx, *J* = 7.1 Hz, 4H), 0.88 (t, *J* = 7.1 Hz, 6H); ^13^C NMR (101 MHz, DMSO-*d*_6_): δ 161.6, 140.2, 128.6, 125.0, 125.0, 123.6, 113.9, 82.5, 46.2, 28.2, 26.8, 22.0, 13.8; ESI-HRMS calcd for [M-I]^+^ C_25_H_31_N_2_S_2_^+^ 423.1923, found 423.1914.*3-Decyl-2-((3-decylbenzo[d]thiazol-2(3H)-ylidene)methyl)benzo[d]thiazol-3-ium iodide* (**11**): using GP2, from 3-decyl-2-methylbenzo[*d*]thiazol-3-ium iodide (417 mg). Yield: 48%; yellow crystals; m.p. 232–233 °C; vis λ_max_ (EtOH): 427 nm, log ε = 4.92; IR λ_max_ (KBr): 3070, 3002, 2920, 2851, 1504, 1465, 1337, 1263, 771, 735, 523 cm^−1^; ^1^H NMR (400 MHz, DMSO-*d*_6_): δ 8.22 (d, *J* = 8.0 Hz, 2H), 7.91 (d, *J* = 8.4 Hz, 2H), 7.69 (t, *J* = 7.5 Hz, 2H), 7.51 (t, *J* = 7.7 Hz, 2H), 6.68 (s, 1H), 4.65 (t, *J* = 7.4 Hz, 4H), 1.77 (p, *J* = 7.3 Hz, 4H), 1.44 (p, *J* = 8.0 Hz, 4H), 1.30 (p, *J* = 7.1 Hz, 4H), 1.27–1.15 (m, 20H), 0.81 (t, *J* = 6.6 Hz, 6H); ^13^C NMR (101 MHz, DMSO-*d*_6_): δ 161.6, 140.2, 128.6, 125.0, 124.9, 123.6, 113.9, 82.6, 46.2, 31.3, 29.0, 29.0, 28.7, 27.1, 26.1, 22.1, 13.9; ESI-HRMS calcd for [M-I]^+^ C_35_H_51_N_2_S_2_^+^ 563.3488, found 563.3476.*3-Methyl-2-((3-methylbenzo[d]oxazol-2(3H)-ylidene)methyl)benzo[d]oxazol-3-ium 4-methylbenzenesulfonate* (**12**): using GP1, from 2-(methylthio)benzoxazole (165 mg), 2-methylbenzoxazole (133 mg), and methyl tosylate (409 mg). Yield: 66%; yellowish white crystals; m.p. 297–298 °C; vis λ_max_ (EtOH): 376 nm, log ε = 4.96; IR λ_max_ (KBr): 3017, 1605, 1597, 1489, 1331, 1220, 1100, 752 cm^−1^; ^1^H NMR (400 MHz, DMSO-*d*_6_): δ 7.83 (d, *J* = 8.0 Hz, 2H), 7.70 (dd, *J* = 8.0, 1.2 Hz, 2H), 7.53 (td, *J* = 7.8, 1.1 Hz, 2H), 7.50–7.40 (m, 4H), 7.10 (d, *J* = 7.8 Hz, 2H), 5.80 (s, 1H), 3.83 (s, 6H), 2.28 (s, 3H); ^13^C NMR (101 MHz, DMSO-*d*_6_): δ 161.8, 146.2, 145.8, 137.5, 131.2, 128.0, 126.0, 125.5, 124.9, 111.3, 111.0, 57.5, 30.7, 20.8.*1-Methyl-2-((1-methylquinolin-2(1H)-ylidene)methyl)quinolin-1-ium iodide* (**13**): using GP3, from 1,2-dimethylquinolin-1-ium 4-methylbenzenesulfonate (329 mg), 1-methyl-2-(methylthio)quinolin-1-ium 4-methylbenzenesulfonate (361 mg) in ethanol. Anion exchange was caused by reflux in a saturated potassium iodide solution (10 mL). Yield: 76%; brown powder; m.p. 240–241 °C (lit. 245–246 °C [34]); vis λ_max_ (EtOH): 523 nm, log ε = 4.60; IR λmax (KBr): 3452, 2981, 1600, 1500, 1328, 1224, 752 cm^−1^; ^1^H NMR (400 MHz, DMSO-*d*_6_): δ 8.14 (d, *J* = 9.3 Hz, 2H), 8.00 (d, *J* = 8.7 Hz, 2H), 7.93 (dd, *J* = 7.9, 1.5 Hz, 2H), 7.83 (ddd, *J* = 8.7, 7.1, 1.6 Hz, 2H), 7.77 (d, *J* = 9.3 Hz, 2H), 7.53 (t, *J* = 7.5 Hz, 2H), 5.83 (s, 1H), 4.00 (s, 6H); ^13^C NMR (101 MHz, DMSO-*d*_6_): δ 154.0, 139.8, 137.5, 132.5, 129.0, 125.0, 124.4, 121.8, 116.8, 92.7, 38.2.*1-Propyl-4-((1-propylquinolin-4(1H)-ylidene)methyl)quinolin-1-ium iodide* (**14**): using GP4, from 4-methyl-1-propylquinolin-1-ium 4-methylbenzenesulfonate (357 mg) and 1-propylquinolin-1-ium 4-methylbenzenesulfonate (343 mg). Anion exchange was made by reflux in a saturated potassium iodide solution (10 mL). Yield: 58%; dark green crystals; m.p. 229–231 °C; vis λ_max_ (EtOH): 593 nm, log ε = 4.69; IR λ_max_ (KBr): 3453, 3011, 2969, 2930, 2872, 1592, 1502, 1491, 1390, 1320, 1218, 1151, 1066, 790, 754, 642, 507 cm^−1^; ^1^H NMR (400 MHz, DMSO-*d*_6_): δ 8.68 (dd, *J* = 8.7, 1.4 Hz, 2H), 8.22 (d, *J* = 7.3 Hz, 2H), 8.00 (dd, *J* = 8.8, 1.1 Hz, 2H), 7.89 (ddd, *J* = 8.6, 6.9, 1.3 Hz, 2H), 7.66 (d, *J* = 7.2 Hz, 2H), 7.63 (ddd, *J* = 7.6, 7.0, 1.1 Hz, 2H), 7.24 (s, 1H), 4.45 (t, *J* = 7.3 Hz, 4H), 1.84 (sx, *J* = 7.4 Hz, 4H), 0.95 (t, *J* = 7.4 Hz, 6H); ^13^C NMR (101 MHz, DMSO-*d*_6_): δ 148.9, 142.9, 137.6, 132.7, 126.1, 126.0, 125.3, 117.6, 108.6, 96.5, 54.8, 22.1, 10.7; ESI-HRMS calcd for [M-I]^+^ C_25_H_27_N_2_^+^ 355.2169, found 355.2162.


#### 3.1.3. General Procedure for the Synthesis of Trimethine Cyanine Dyes **15** and **16**


A solution of 2-methylbenzoazolium salt (1.0 mmol) and triethyl orthoformate (2.1 mmol, 349 µL) in 4 mL of dry pyridine was refluxed for 5 to 48 h. After cooling to room temperature, diethyl ether was added, and the resulting mixture was refrigerated to allow for complete precipitation. The crystalline product was collected via filtration under reduced pressure, washed with diethyl ether, and recrystallised from dry acetonitrile or methanol [36,37].

3-Ethyl-2-(3-(3-ethylbenzo[*d*]oxazol-2(3*H*)-ylidene)prop-1-en-1-yl)benzo[*d*]oxazol-3-ium iodide (**15**): from 3-ethyl-2-methylbenzo[*d*]oxazol-3-ium iodide (289 mg) for 48 h. Yield: 24%; red solid; m.p. 281–282 °C (lit. 277–279 °C dec. [41]); vis λ_max_ (EtOH): 485 nm, log ε = 5.15; IR λmax (KBr): 2975, 1565, 1508, 1458, 1449, 1398, 1365, 1349, 1206, 1113, 1085, 968, 916, 735 cm^−1^; ^1^H NMR (400 MHz, DMSO-*d*_6_): δ 8.32 (t, *J* = 13.3 Hz, 1H), 7.77 (d, *J* = 8.0 Hz, 2H), 7.71 (d, *J* = 7.8 Hz, 2H), 7.48 (t, *J* = 7.4 Hz, 2H), 7.41 (t, *J* = 7.8 Hz, 2H), 6.09 (d, *J* = 13.3 Hz, 2H), 4.26 (q, *J* = 7.2 Hz, 4H), 1.37 (t, *J* = 7.2 Hz, 6H); ^13^C NMR (101 MHz, DMSO-*d*_6_): δ 161.4, 146.5, 146.2, 130.9, 125.9, 125.0, 111.2, 110.9, 84.9, 39.0, 12.8.

3-Ethyl-2-(3-(3-ethylbenzo[*d*][1,3]selenazol-2(3*H*)-ylidene)prop-1-en-1-yl)benzo[*d*][1,3]selenazol-3-ium iodide (**16**): from 3-ethyl-2-methylbenzo[*d*][1,3]selenazol-3-ium iodide (352 mg) for 5 h. Yield: 95%; dark green crystals; m.p. 272–273 °C (lit. 270–271 °C dec. [42]); vis λ_max_ (EtOH): 573 nm, log ε = 4.94; IR λ_max_ (KBr): 3056, 2969, 2870, 1585, 1548, 1454, 1416, 1317, 1262, 1125, 1020, 927, 822, 774, 740 cm^−1^; ^1^H NMR (400 MHz, DMSO-*d*_6_): δ 8.10 (dd, *J* = 8.0, 1.3 Hz, 2H), 7.81 (t, *J* = 12.4 Hz, 1H), 7.72 (d, *J* = 8.4 Hz, 2H), 7.56 (ddd, *J* = 8.2, 7.3, 1.2 Hz, 2H), 7.37 (t, *J* = 7.9 Hz, 2H), 6.79 (d, *J* = 12.4 Hz, 2H), 4.38 (q, *J* = 7.1 Hz, 4H), 1.34 (t, *J* = 7.1 Hz, 6H); ^13^C NMR (101 MHz, DMSO-*d*_6_): δ 168.8, 151.7, 142.3, 128.0, 126.5, 125.4, 125.1, 114.7, 103.2, 42.3, 12.6.

#### 3.1.4. General Procedure for the Synthesis of Heptamethine Cyanine Dyes **17**–**19**

A solution of 2-methylbenzoazolium salt (1.0 mmol) and 2-chloro-1-formyl-3-(hydroxymethylene)cyclohex-1-ene (1.5 mmol, 259 mg) in anhydrous butan-1-ol/benzene (7:3) (70 mL) was heated under reflux over activated 4Å molecular sieves (1000% by mass of benzoazolium salt) until complete consumption of the benzoazolium salt (10–15 min). Then, an additional molar equivalent of 2-methylbenzoazolium salt and dry pyridine (14 mL) was added, and the resulting mixture was refluxed for 30 to 40 min. After the separation of the molecular sieves by filtration, the chloro dyes **17-19** precipitated from the cooled reaction mixture directly following partial solvent removal or by the addition of diethyl ether. The solid so obtained was collected by filtration under reduced pressure, washed with water and diethyl ether, and recrystallised from methanol or ethanol [38,73]. 


*2-(2-(2-Chloro-3-(2-(3-pentylbenzo[d]thiazol-2(3H)-ylidene)ethylidene)cyclohex-1-en-1-yl)vinyl)-3-pentylbenzo[d]thiazol-3-ium iodide* (**17**): from 2-methyl-3-pentylbenzo[*d*]thiazol-3-ium iodide (2 × 347 mg). Yield: 56%; green crystals; m.p. 227–228 °C; vis λ_max_ (EtOH): 801 nm, log ε = 5.45; IR λ_max_ (KBr): 3058, 3000, 2925, 2859, 1579, 1530, 1503, 1459, 1428, 1403 cm^−1^; ^1^H NMR (400 MHz, DMSO-*d*_6_): δ 7.95 (d, *J* = 7.9 Hz, 2H), 7.78 (d, *J* = 13.4 Hz, 2H), 7.72 (d, *J* = 8.3 Hz, 2H), 7.54 (t, *J* = 7.8 Hz, 2H), 7.36 (t, *J* = 7.7 Hz, 2H), 6.45 (d, *J* = 13.5 Hz, 2H), 4.39 (t, *J* = 6.8 Hz, 4H), 2.65 (t, *J* = 6.2 Hz, 4H), 1.83 (q, *J* = 5.5 Hz, 2H), 1.71 (p, *J* = 6.9 Hz, 4H), 1.46–1.26 (m, 8H), 0.87 (t, *J* = 6.6 Hz, 6H); ^13^C NMR (101 MHz, DMSO-*d*_6_): δ 163.4, 144.2, 141.6, 140.5, 128.2, 125.4, 125.2, 124.3, 123.1, 113.6, 100.0, 46.1, 28.1, 27.1, 26.5, 21.8, 20.4, 13.9.*2-(2-(2-Chloro-3-(2-(3-decylbenzo[d][1,3]selenazol-2(3H)-ylidene)ethylidene)cyclohex-1-en-1-yl)vinyl)-3-decylbenzo[d][1,3]selenazol-3-ium iodide* (**18**): from 3-decyl-2-methylbenzoselenazol-3-ium iodide (2 × 464 mg). Yield: 44%; green crystals; m.p. 221–224 °C; vis λ_max_ (EtOH): 814 nm, log ε = 5.46; IR λ_max_ (KBr): 3060, 2921, 2850, 1660, 1578, 1524, 1503, 1450, 1428, 1399 cm^−1^; ^1^H NMR (400 MHz, DMSO-*d*_6_): δ 8.04 (d, *J* = 7.9 Hz, 2H), 7.70 (d, *J* = 10.2 Hz, 4H), 7.59–7.49 (m, 2H), 7.35 (t, *J* = 7.6 Hz, 2H), 6.60 (d, *J* = 13.2 Hz, 2H), 4.42 (t, *J* = 6.9 Hz, 4H), 2.73–2.62 (m, 4H), 1.91–1.79 (m, 2H), 1.71 (p, *J* = 6.9 Hz, 4H), 1.45–1.16 (m, 28H), 0.84 (t, *J* = 6.9 Hz, 6H); ^13^C NMR (101 MHz, DMSO-*d*_6_): δ 168.5, 144.1, 143.1, 142.3, 128.2, 126.4, 125.4, 125.2, 125.1, 115.1, 103.8, 46.9, 31.3, 28.9, 28.7, 28.6, 27.2, 26.7, 25.8, 22.1, 14.0.*2-(2-(2-Chloro-3-(2-(3,3-dimethyl-1-(4-sulfobutyl)indolin-2-ylidene)ethylidene)cyclohex-1-en-1-yl)vinyl)-3,3-dimethyl-1-(4-sulfobutyl)-3H-indol-1-ium iodide* (**19-IR-783**): from 2,3,3-trimethyl-1-(4-sulfobutyl)-3*H*-indol-1-ium iodide (2 × 423 mg). Yield: 65%; blue crystals; m.p. 220–225 °C; vis λ_max_ (EtOH): 787 nm, log ε = 4.98; IR λ_max_ (KBr): 3052, 2912, 2853, 1729, 1549, 1450, 1304, 1253, 1177, 1087, 975 cm^−1^; ^1^H NMR (400 MHz, DMSO-*d*_6_): δ 8.26 (d, *J* = 14.1 Hz, 2H), 7.99 (s, 2H), 7.62 (d, *J* = 7.4 Hz, 2H), 7.49 (d, *J* = 7.9 Hz, 2H), 7.42 (t, *J* = 7.7 Hz, 2H), 7.28 (t, *J* = 7.4 Hz, 2H), 6.37 (d, *J* = 14.1 Hz, 2H), 4.22 (t, *J* = 7.4 Hz, 4H), 2.73 (t, *J* = 6.1 Hz, 4H), 1.90–1.77 (m, 8H), 1.77–1.69 (m, 6H), 1.67 (s, 12H); ^13^C NMR (101 MHz, DMSO-*d*_6_): δ 172.1, 148.0, 143.1, 142.1, 141.1, 128.7, 126.3, 125.1, 122.5, 111.7, 101.8, 50.7, 49.0, 43.8, 36.5, 27.5, 26.1, 25.9, 22.5.


### 3.2. In Vitro Studies

For the in vitro studies, a dimethyl sulfoxide stock solution at the concentration of 1 mM was prepared for each compound under investigation. These solutions were stored at 4 °C in an environment protected from light.

#### 3.2.1. Antiproliferative Effects in Human Cell Lines

##### Cell Culture

Caco-2, MCF-7, PC-3, and NHDF cell lines were obtained from the American Type Culture Collection (ATCC) and maintained in 75 cm^2^ culture flasks in a humidified atmosphere at 37 °C with 5% carbon dioxide using a LEEC Culture Safe Precision CO_2_ Incubator. The culture media was renewed every two to three days. The NHDF cell line was cultured in Roswell Park Memorial Institute (RPMI) medium supplemented with 10% of inactivated fetal bovine serum (FBS), 2 mM of L-glutamine, 10 mM of 4-(2-hydroxyethyl)-1-piperazineethanesulfonic acid (HEPES), 1 mM of sodium pyruvate, and 1% of antibiotic/antimycotic mixture (Ab; 10,000 U/mL of penicillin G, 100 mg/mL of streptomycin, and 25 µg/mL of amphotericin B). The PC-3 cell line was also cultured in RPMI 1640 medium supplemented with 10% inactivated FBS and 1% of an antibiotic mixture (Sp; 10,000 U/mL of penicillin G and 100 mg/mL of streptomycin). The Caco-2 and MCF-7 cell lines were grown in Dulbecco’s Modified Eagle Medium (DMEM). Caco-2 was supplemented with 20% of inactivated FBS and 1% of Sp. On the other hand, the MCF-7 medium was complemented with 10% of inactivated FBS and 1% of Ab. All manipulations were performed in a Labculture Class II A2 Biological Safety Cabinet.

##### Cell Viability Using an MTT Assay

After the near confluency of cultures, cells were detached with a solution of 0.125 g/L trypsin and 0.02 g/L ethylenediaminetetracetic acid (EDTA), counted using the trypan blue method, resuspended, and seeded in 96-well culture plates (100 µL, 2 × 10^4^ cells/mL). Following 48 h of adherence, cells were treated with the test solutions and incubated for 72 h. This incubation, as well as the solution manipulations, were performed in an environment protected from light. After the incubation, the medium was replaced with a fresh incomplete culture medium (without FBS and Ab or Sp) and MTT solution (5 mg/mL in phosphate buffer saline (PBS)). After additional incubation for 4 h, the MTT medium was removed, formazan crystals were dissolved in DMSO, and the optical density of each well was measured at 570 nm using a microplate spectrophotometer Bio-Rad xMark. Untreated cells were used as a negative control and 5-FU as the positive control. The results were expressed as the relative cell proliferation relative to the negative control cells. At least two independent assays in quadruplicate were achieved for each cell viability study. To determine the IC_50_ values, Caco-2, MCF-7, PC-3, and NHDF cells were exposed to increasing concentrations between 0.001 and 100 μM.

##### Photocytotoxicity Evaluation

The photocytotoxicity evaluation was performed on the NHDF cell line similar to the above MTT cell viability assay. In contrast to 72 h continuous incubation in the dark with test solutions, cells were subjected to 30 min of constant irradiation with white light from an RGBW LED projector (220–240 V and 30 W, Luxtar) after 48 h of incubation. This light source emits radiation at wavelengths suitable for the excitation of evaluated dye (see support information) with a total fluence rate of 1.196 W/sr·m^2^ (measured with a pr-650 spectrascan spectroradiometer, Photo Research Inc., Chatsworth, CA, USA). During the photocytotoxicity treatment, the device was placed over the culture cell plates supported by a closed system at a distance of 12 cm. The maximum temperature of the exposure of the cells was 37 °C. After this irradiation, cells were incubated for an additional 24 h, making up a total of 72 h incubation and exposure to the test solutions. To check the validity of the light source, the squaraine dye **4**, which was previously described with relevant photodynamic activity [21], was used as a positive control. A photocytotoxicity assay was performed in quadruplicate, and at least two independent assays were achieved.

##### Photostability Evaluation

The photostability of the dye **12** was evaluated using a concentration of 10 µM prepared from the dilution of 10 mM DMSO stock solution in PBS. In a 96-well plate, 100 µL of dye solution was added, irradiated in the same way as that previously described for photocytotoxicity evaluation for 120 min, and spectrophotometric readings were taken over this period at the dye **12**’s maximum absorption wavelength (376 nm). Two independent experiments were performed in quadruplicate. The absorbance was normalised to simplify the analysis.

##### Morphological Analysis

After the near confluency of the culture, trypsinisation, and counting, Caco-2 cells were seeded (3 mL; 3 × 10^4^ cells/mL) in 35 mm cell culture dishes with treatment for the tissue culture. Following 48 h of adherence, the cells were treated with the test solutions and incubated for 72 h in an environment protected from light. At every 24 h, cellular morphology was verified in an Olympus CKX41 SF inverted microscope with a 10, 20, or 40× objective, and the analysis was based on comparing the cells with and without treatment. Cell imaging was captured using a DC6V Olympus digital camera coupled to the microscope.

#### 3.2.2. Flow Cytometry

The apoptosis occurrence and influence of the dye **12** on the cell cycle were evaluated using propidium iodide (PI) staining and flow cytometry, as described by Riccardi and Nicolleti [54]. After the near confluency of the culture, trypsinisation, and counting, Caco-2 cells were seeded (2 mL, 3 × 10^4^ cells/mL) in 6-well plates. After 48 h of adherence, the cells were treated with the test solutions (3 mL and 5 mL for negative control) and incubated for 72 h. Untreated cells were used as the negative control, and sorbitol [55] and 5-FU [56] were used as positive controls for the apoptosis and cell cycle distribution studies, respectively. After incubation, the supernatant was collected and the cells were washed with PBS and combined with the cells harvested using the trypsin treatment. This cell suspension was maintained on ice and centrifugated, and then the pellet was resuspended in 0.5 mL of PBS. The cells were fixed by adding 4.5 mL of 70% ethanol cooled at −20 °C and were stored at the same temperature. For the cytometry analysis, PBS was added to the fixed cells before centrifugation, and then the ethanolic supernatant was removed. The pellet was resuspended in 0.5 mL of PBS and 0.5 mL of DNA extraction (192 mL of 0.2 M sodium phosphate and 8 mL of 0.1% Triton X-100, pH 7.8) was added. After incubation for 5 min, the cells were centrifuged, the supernatant was removed, and the pellet resuspended in 0.3 mL of staining solution (20 µg/mL of PI and 0.2 mg/mL of DNase-free RNase in PBS). Using a FACSCalibur flow cytometer and the 488 nm laser line, a minimum of 20,000 events were recorded. Acquisition and apoptosis analysis was performed using BD CellQuestTM Pro 4.0.2 Software. Cell cycle analysis was performed with ModFit LT 5.0.9 software. To exclude debris and necrotic cells, events with a lower diameter (forward scatter; FCS) and reduced PI fluorescence (FL3) than the hypodiploid apoptotic cells were eliminated. Two independent experiments were performed.

### 3.3. Computational Studies

In order to predict the physicochemical properties, druglikeness, and medicinal chemistry principles of the dye **12**, an in silico study was performed using the online tool SwissADME [58].

## 4. Conclusions

A series of representative mono-, tri-, and heptamethine cyanine dyes containing benzothiazole, benzoxazole, benzoselenazole, indole, and quinoline as terminal nitrogen heterocycles; methyl, ethyl, ethylenecarboxymethyl, pentyl, benzyl, and dodecyl as *N*-alkyl chains; and chloride, bromide, iodide, perchlorate, and tosylate as counterions was successfully synthesised using or based on classical methods. These dyes were further fully characterised, and their antiproliferative potentials in Caco-2, MCF-7, and PC-3 human tumour cell lines and in a non-tumourous cell line NHDF were investigated while envisioning their use as anti-cancer agents. The influence of structural variations on their biological activity was analysed, leading us to conclude that within the cyanine study set, the benzoxazole monomethine dye was more selective for cancerous PC-3, MCF-7, and Caco-2 cell lines over the NHDF cell lines. While not so decisive, the decrease in the *N*-substituent chain led to an increase in both the cytotoxicity and SI, where the nature of counterion had some influence over both the cytotoxicity and SI between the cancerous Caco-2 and non-cancerous NHDF. From all these results, a monomethine benzoxazole cyanine dye was elected as the best one while considering the best compromise between cytotoxicity and selectivity (IC_50_ for Caco-2 = 0.67 µM, SI = 20.9). Further studies on this dye revealed an absence of photocytoxicity and approximately 4% degradation after 30 min of light irradiation. The morphological analyses of the Caco-2 cells layer revealed that exposure to 1 or 10 µM of the referred dye after 24, 48, and 72 h led to morphological changes compared with the negative control cells. After 72 h of treatment, a notorious increase in the number of cells in the G0/G1 phase and a decrease in the S phase compared with untreated Caco-2 cells were also observed despite the absence of apoptosis induction. In silico studies demonstrated that this dye fulfiled the main druglikeness filters, and good intestinal absorption and oral bioavailability were expected.

The residual photodegradation or even absence in internal tissues, the lack of photodynamic action, and the remarkable values of cytotoxicity and selective index found for the selected cyanine dye in this study seemed to clearly point to the potential use of this family of cyanine dyes as anti-proliferative agents.

## Data Availability

Not applicable.

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
