# Peer review of "An Insight into Symmetrical Cyanine Dyes as Promising Selective Antiproliferative Agents in Caco-2 Colorectal Cancer Cells"

_molecules, 2022, doi:10.3390/molecules27185779_

Round 1

Reviewer 1 Report

The authors reported synthesizing and evaluating eighteen symmetric mono, tri, and heptamethine cyanine dyes as potential anticancer agents.

The described synthesis is not innovative, some of the obtained compounds were described earlier, the described studies are not clear, and obtained results are not too promising. Thus the manuscript requires important corrections.

I present my comments below.

1) Line 98: the authors wrote “ monomethine cyanines 4-11 and 14 were never described” – and they give reference for compound 4 in table 1– thus, was this compound described earlier or not?

2) in section “2.2.1. Antiproliferative effects in human cell lines”

– why the tests were carried out on a single concentration of 10 μM? How was this concentration determined? An explanation is required.

- Figure 1 is not clear, and is not legible. The interpretation of obtained values is impossible. Figure 1 requires correction.

- Figure 2 - in the description of the figure, the authors give the concentration of 10 µM for dye 12, while the figure shows the increasing concentration of the dye 12 – correction/explanation required

- compounds 15 and 16 also showed interesting SI values, why extended tests were not performed for them as for compound 12

3) in section “2.2.3. Morphological analysis of Caco-2 cells” - Why were the tests performed for the concentrations of 1 μM and 10 μM for 12 and 10 μM for 5-FU? Why was the IC50 value not used?

3) in section “2.2.4. Flow cytometry studies of apoptosis and cell cycle effects”- Why were the tests performed for the concentration of 10 μM ? Why was the IC50 value not used?

IC50 values are used in investigating the mechanism of compounds and this investigation should be added.

4) in the description of NMR spectra “ppm” is not required – appropriate information is in section “3.1. Chemistry: “..chemical shift (δ) values are given in parts per million (ppm)”

5) line 523-524 – why two methods were used? Why was not selected method with better yield?

6) in “Conclusions” authors wrote: "The influence of structural variations on their biological activity was analyzed, leading us to conclude that benzoxazole monomethine dyes appear to be selective for cancerous PC-3, MCF-7, and Caco-2 cell lines concerning NHDF cell lines.

This conclusion, derived only from the single concentration study, is premature and too far-fetched. More extensive and in-depth studies are needed.

Author Response

Reviewer 1:

The authors reported synthesizing and evaluating eighteen symmetric mono, tri, and heptamethine cyanine dyes as potential anticancer agents.

The described synthesis is not innovative, some of the obtained compounds were described earlier, the described studies are not clear, and obtained results are not too promising. Thus the manuscript requires important corrections.

I present my comments below.

1) Line 98: the authors wrote “monomethine cyanines 4-11 and 14 were never described” – and they give reference for compound 4 in table 1– thus, was this compound described earlier or not?

Thank you very much for your attention call. The change to “monomethine cyanines 8-11 and 14 were never described” was made accordingly. This statement has been verified once more and to the best of our knowledge it was correct.

2) in section “2.2.1. Antiproliferative effects in human cell lines”

– why the tests were carried out on a single concentration of 10 μM? How was this concentration determined? An explanation is required.

A single-concentration screening assay was performed based on other authors work (e.g. Molecules 2018, 23(3), 574; Eur.J.Med.Chem. 204 (2020) 112556) and goes in line with Development Therapeutic Program (DTP) of National Cancer Institute (NCI) Screening Methodology. Accordingly, only compounds which exhibit significant cell growth inhibition in the screening are evaluated at various concentrations, which allows a significant reduction of spent time and associated costs. Despite that different concentrations of the compounds can be found in cytotoxicity screenings (e.g. 10 µM, 20 µM, 30 µM, …) (e.g. Molecules 2018, 23(3), 574; Eur.J.Med.Chem. 204 (2020) 112556, 284-291; Eur.J.Med.Chem., 2018, 143, 829-842), in this research work we have chosen 10 µM. The reason for this choice is that this concentration is near or higher than the maximum therapeutic plasmatic levels achieved with the majority of drugs (Applied Biopharmaceutics & Pharmacokinetics, 7th edition, 2016, Leon Shargel, Andrew B.C. Yu, McGrawHill; Clinical Pharmacokinetis, 6th edition, 2017, John E. Murphy, ASHP publications) and therefore can be informative on the decision of continue or not studying the antiproliferative effects of compounds under development as potential anticancer drugs.

- Figure 1 is not clear, and is not legible. The interpretation of obtained values is impossible. Figure 1 requires correction.

Table S2 containing all data of 10 µM screening was added in supplementary material. Nevertheless, if suitable and for a better interpretation of the values, a second graph showing the data between 0-20% can be added to figure 1.

- Figure 2 - in the description of the figure, the authors give the concentration of 10 µM for dye 12, while the figure shows the increasing concentration of the dye 12 – correction/explanation required

Thank you for attention call. The description of Figure 2 was erroneous. Correction was made accordingly.

- compounds 15 and 16 also showed interesting SI values, why extended tests were not performed for them as for compound 12

In our work, we considered that SI is more relevant than the potency by itself, despite 15 and 16 also showed interesting SI values. This choice was clarified in the last paragraph of section 2.2.1, namely: “Although trimethine cyanine dyes 15-16 showed to be more potent in Caco-2 cells than monomethine cyanine dye 12, the last one showed a best compromise between potency against the Caco-2 cell line and selectivity for cancer concerning non-cancer cells. In conclusion, and looking in a global sense to all the results presented above (Table 2), the monomethine cyanine dye 12 is here appointed as the most promising compound under this study and was selected for further studies.”

3) in section “2.2.3. Morphological analysis of Caco-2 cells” - Why were the tests performed for the concentrations of 1 μM and 10 μM for 12 and 10 μM for 5-FU? Why was the IC50 value not used?

3) in section “2.2.4. Flow cytometry studies of apoptosis and cell cycle effects”- Why were the tests performed for the concentration of 10 μM ? Why was the IC50 value not used?

IC50 values are used in investigating the mechanism of compounds and this investigation should be added.

In a first instance, morphological analysis was performed at several concentrations prior to the presented studies, including at the concentration of 0.5 μM (close to the IC50 value). However, after 72h of incubation, a high number of cells were present, being difficult to identify the morphological differences caused by the cyanine dye, as well as by 5-FU at 1 μM. On the other hand, Caco-2 cells tend to form a high number of cell aggregates (as visible at Figures 5 and 6). In fact, even after a prolonged pellet resuspended at the final step of flow cytometry, the untreated stained cells suspension had to be filtered before been analysed on the cytometer. To overcome this issue, we ensured that the final number of cells treated was close to necessary to perform the analysis. Thus, we reach to the 10 μM for 5-FU and dye 12, overcoming the last step of filtration and therefore reducing the errors associated with the assay.

4) in the description of NMR spectra “ppm” is not required – appropriate information is in section “3.1. Chemistry: “..chemical shift (δ) values are given in parts per million (ppm)”

Thank you very much for observation. Changed accordingly.

5) line 523-524 – why two methods were used? Why was not selected method with better yield?

Thank you very much for observation. The HCl method was removed accordingly.

6) in “Conclusions” authors wrote: "The influence of structural variations on their biological activity was analyzed, leading us to conclude that benzoxazole monomethine dyes appear to be selective for cancerous PC-3, MCF-7, and Caco-2 cell lines concerning NHDF cell lines.

This conclusion, derived only from the single concentration study, is premature and too far-fetched. More extensive and in-depth studies are needed.

Thank you very much for observation. However, this conclusion was based on IC50 studies performed in all cell lines determined using several concentrations present at table 2. Nevertheless, we changed this conclusion to the following less far-fetched statement “The influence of structural variations on their biological activity was analyzed, leading us to conclude that within the cyanine study set, the benzoxazole monomethine dye was more selective for cancerous PC-3, MCF-7, and Caco-2 cell lines concerning NHDF cell lines”.

Reviewer 2 Report

The manuscript describes the synthesis and evaluation of eighteen symmetric mono, tri, and heptamethine cyanine dyes as potential anticancer agents. The influence of heterocycle nature, counterion, and methine chain length on antiproliferative effect and selectivity was analysed, and relevant structure-activity relationships data were achieved. Most of the monomethine and trimethine cyanine dyes under study demonstrated a good antiproliferative effect on a panel of three different cancer human cell lines Among the synthesized compounds, the monomethine cyanine dye derived from benzoxazole, showed IC50 for Caco-2 = 0.67 μM and selectivity index of 20.9 for Caco-2 versus normal human dermal fibroblasts - NHDF), led to a Caco-2 cell cycle arrest at G0/G1. The manuscript should be revised following these comments:

The title should be revised, the tested compounds should be also evaluated against other colorectal cancer cell lines such as HT-29 and SW620.

Self-citations should be reduced  (16 references on a total of 80)

All synthesized compounds were characterized by the absence of substituents on the benzene portion of benzoheterocycle ring. What’s the rationale?

The target of the most active compound and its effect on the depolarization of mitochondrial membrane potential.

Morphological analysis of Caco-2 cells, flow cytometry studies of apoptosis and cell cycle effects were analysed at the concentration of 10 micromolar. What is the effect at an exposure corresponding to IC50 value?

Author Response

Reviewer 2:

The manuscript describes the synthesis and evaluation of eighteen symmetric mono, tri, and heptamethine cyanine dyes as potential anticancer agents. The influence of heterocycle nature, counterion, and methine chain length on antiproliferative effect and selectivity was analysed, and relevant structure-activity relationships data were achieved. Most of the monomethine and trimethine cyanine dyes under study demonstrated a good antiproliferative effect on a panel of three different cancer human cell lines Among the synthesized compounds, the monomethine cyanine dye derived from benzoxazole, showed IC50 for Caco-2 = 0.67 μM and selectivity index of 20.9 for Caco-2 versus normal human dermal fibroblasts - NHDF), led to a Caco-2 cell cycle arrest at G0/G1. The manuscript should be revised following these comments:

The title should be revised, the tested compounds should be also evaluated against other colorectal cancer cell lines such as HT-29 and SW620.

Thank you for the observation. The title was revised to “An insight into symmetrical cyanine dyes as promising selective antiproliferative agents in Caco-2 colorectal cancer cells”

Self-citations should be reduced  (16 references on a total of 80)

Self-citations were half reduced accordingly.

All synthesized compounds were characterized by the absence of substituents on the benzene portion of benzoheterocycle ring. What’s the rationale?

The original goal of this work was to explore the best heterocycle nature, counterion, N-substituent and methine chain length on antiproliferative effect and selectivity index. Based on the best result herein presented (dye 12 with a benzoxazole moiety, tosylate counterion, methyl N-substituent and methine chain with n=0 or 1), further studies will be engaged to analyse the influence of substituents on the benzene portion. Due to the almost infinite hypothesis of substituents to introduce, this study is just beginning.

The target of the most active compound and its effect on the depolarization of mitochondrial membrane potential.

Morphological analysis of Caco-2 cells, flow cytometry studies of apoptosis and cell cycle effects were analysed at the concentration of 10 micromolar. What is the effect at an exposure corresponding to IC50 value?

In a first instance, morphological analysis was performed at several concentrations prior to the presented studies, including at the concentration of 0.5 μM (close to the IC50 value). However, after 72h of incubation, a high number of cells were present, being difficult to identify the morphological differences caused by the cyanine dye, as well as by 5-FU at 1 μM. On the other hand, Caco-2 cells tend to form a high number of cell aggregates (as visible at Figures 5 and 6). In fact, even after a prolonged pellet resuspended at the final step of flow cytometry, the untreated stained cells suspension had to be filtered before been analysed on the cytometer. To overcome this issue, we ensured that the final number of cells treated was close to necessary to perform the analysis. Thus, we reach to the 10 μM for 5-FU and dye 12, overcoming the last step of filtration and therefore reducing the errors associated with the assay.

Reviewer 3 Report

In this manuscript, Almeida and co-workers are reporting an interesting application of the symmetric cyanine dyes as anti-proliferative agents towards colorectal cancer. The overall report of this work seems to be interesting and potentially useful to the field. Therefore, I recommend accepting this article after a revision. Following concerns must be addresses appropriately.

(1). There are significant grammatical and textual errors and inconsistencies in the manuscript. This should be thoroughly revised.

(2). Authors should summarize photophysical properties of the reporting dyes in different solvents.

(3). All synthetic schemes and procedures should show reaction yields.

(4). Why only assays conducted is 10 uM concentration? This seems to be incomplete and unjustifiable. These assays must be conducted at least 3-5 different concentrations in order to validate the outcomes.

(5). Figure 1 representation is not appropriate, and it looks overloaded. Authors must attempt to represent this figure in a more diluted and a readable format.

(6). Authors should clearly discuss the outcomes of the photostability evaluation data by considering chemical structures of the dyes.

(7). Authors should provide HRMS data for all compounds and must include under the methods section with ppm error.

(8). Conclusions section must be revised properly to clearly deliver the significant outcomes of this work. The current format is lacking significance.

Author Response

Reviewer 3:

In this manuscript, Almeida and co-workers are reporting an interesting application of the symmetric cyanine dyes as anti-proliferative agents towards colorectal cancer. The overall report of this work seems to be interesting and potentially useful to the field. Therefore, I recommend accepting this article after a revision. Following concerns must be addresses appropriately.

(1). There are significant grammatical and textual errors and inconsistencies in the manuscript. This should be thoroughly revised.

Text was revised by an English expert.

(2). Authors should summarize photophysical properties of the reporting dyes in different solvents.

Due to the low solubility of cyanine dyes, photophysical properties as absorption spectra were generally made in ethanol. The absorptivity molar values were additionally added to the wavelength of maximum absorption on table 1.

(3). All synthetic schemes and procedures should show reaction yields.

Thank you very much for observation. Changed accordingly.

(4). Why only assays conducted is 10 uM concentration? This seems to be incomplete and unjustifiable. These assays must be conducted at least 3-5 different concentrations in order to validate the outcomes.

A single-concentration screening assay was performed based on other authors work (e.g. Molecules 2018, 23(3), 574; Eur.J.Med.Chem. 204 (2020) 112556) and goes in line with Development Therapeutic Program (DTP) of National Cancer Institute (NCI) Screening Methodology. Accordingly, only compounds which exhibit significant cell growth inhibition in the screening are evaluated at various concentrations, which allows a significant reduction of spent time and associated costs. Despite that different concentrations of the compounds can be found in cytotoxicity screenings (e.g. 10 µM, 20 µM, 30 µM, …) (e.g. Molecules 2018, 23(3), 574; Eur.J.Med.Chem. 204 (2020) 112556, 284-291; Eur.J.Med.Chem., 2018, 143, 829-842), in this research work we have chosen 10 µM. The reason for this choice is that this concentration is near or higher than the maximum therapeutic plasmatic levels achieved with the majority of drugs (Applied Biopharmaceutics & Pharmacokinetics, 7th edition, 2016, Leon Shargel, Andrew B.C. Yu, McGrawHill; Clinical Pharmacokinetis, 6th edition, 2017, John E. Murphy, ASHP publications) and therefore can be informative on the decision of continue or not studying the antiproliferative effects of compounds under development as potential anticancer drugs. Besides, IC50 values presented in figure 2 and table 2 were calculated from a serial of six different concentrations of at least two independent assays performed in quadruplicate.

(5). Figure 1 representation is not appropriate, and it looks overloaded. Authors must attempt to represent this figure in a more diluted and a readable format.

Table S2 containing all data of 10 µM screening was added in supplementary material. Nevertheless, if suitable and for a better interpretation of the values, a second graph showing the data between 0-20% can be added to figure 1.

(6). Authors should clearly discuss the outcomes of the photostability evaluation data by considering chemical structures of the dyes.

Thank you very much for observation. The following sentence was added in the last paragraph of section 2.2.2: “However, the expectable absence of light in an eventual use of monomethine cyanine dyes for treatment of internal cancer cells tissues may prevent the residual photodegradation of these dyes. Moreover, the absence of any photodynamic action prevents the most common side effect of PDT, namely the sensitivity to bright lights and sunlight, wich is an additional advantage to the potential use of these family of cyanine dyes as anti-proliferative agents.”

(7). Authors should provide HRMS data for all compounds and must include under the methods section with ppm error.

Accordingly to Molecules author instructions, “Reports on previously undescribed organic compounds should include, as supplementary data, 1H, 13C and/or other key heteronuclear or 2D NMR spectra, together with High Resolution Mass Spectrometry (HRMS) or elemental analysis.” Therefore, the HRMS spectra were presented just for all undescribed organic compounds, in the format of ESI-HRMS Calcd and found values (ppm error).

Besides and accordingly to author instructions, the High Resolution Mass Spectra were additionally presented as supplementary data.

(8). Conclusions section must be revised properly to clearly deliver the significant outcomes of this work. The current format is lacking significance.

Thank you very much for your suggestion. The conclusion section was revised accordingly. The following paragraph was added to deliver some expected outcomes of this work: “The residual photodegradation or even absence in internal tissues, the lack of photodynamic action and the remarkable values of cytotoxicity and selective index found for the selected cyanine dye on this study, seems to clear point to the potential use of these family of cyanine dyes as anti-proliferative agents.”

Round 2

Reviewer 1 Report

I accept in present form.

Reviewer 2 Report

The manuscript deserves to be published in the present form. The authors  have answered to all my comments. 

Reviewer 3 Report

Although, Authors have not fully answered my concerns for this manuscript I raised previously, I can see they are providing some explanations based on previously published research.

I am nor fully convince with that solubility of this cyanine dyes will be a significant facto. Especially if they have such solubility issues, the assay results such as MTT will also be doubtful as well.

However, the application of these probes is mainly focusing on therapeutics and not for bioimaging, I will not pursue this issue any further.

There are still few typo and grammatical errors please check carefully  prior to the final acceptance.